# Core versus Surface Sensors for Reinforced Concrete Structures: A Comparison of Fiber-Optic Strain Sensing to Conventional Instrumentation

**DOI:** 10.3390/s23031745

**Published:** 2023-02-03

**Authors:** Ryan Hoult, Alex Bertholet, João Pacheco de Almeida

**Affiliations:** Institute of Mechanics, Materials and Civil Engineering, Université Catholique de Louvain, 1348 Louvain-la-Neuve, Belgium

**Keywords:** fiber optic, DFOS, DOFS, optical, RC, DIC, yield, strain penetration, walls

## Abstract

High-resolution distributed reinforcement strain measurements can provide invaluable information for developing and evaluating numerical and analytical models of reinforced concrete structures. A recent testing campaign conducted at UCLouvain in Belgium used fiber-optic sensors embedded along several longitudinal steel rebars of three reinforced concrete U-shaped walls. The resulting experimental dataset provides an opportunity to evaluate and compare, for different types of loading, the strain measurements obtained with the fiber-optic sensors in the confined core of the structural member against more conventional and state-of-the-practice sensors that monitor surface displacements and deformations. This work highlights the need to average strain measurements from digital image correlation techniques in order to obtain coherent results with the strains measured from fiber optics, and investigates proposals to achieve this relevant goal for research and engineering practices. The longitudinal strains measured by the fiber optics also provide additional detailed information on the behavior of these wall units compared to the more conventional instrumentation, such as strain penetration into the foundation and head of the wall units, which are studied in detail.

## 1. Introduction

The ability to measure the entire rebar strain profile within reinforced concrete (RC) structures has, besides direct applications in short- or long-term structural health monitoring, the potential to solve many long-standing questions on their mechanical response by helping to characterize and quantify different phenomena that are currently only known qualitatively. For example, previous works on tension stiffening [1] and concrete cracking [2] could not have been conducted without deriving a partially complete reinforcement strain profile along the steel reinforcing bars (commonly referred to as “rebars”). However, up until recently, strain gauges were used along the length of the reinforcement, a time-consuming and costly exercise [3]; take the experimental study by Scott and Gill [1], for example, which required 84 strain gauges to be installed within a single longitudinal reinforcement bar, or the pioneering and well-known investigations by Shima et al. [4] for the development of local bond stress-slip-strain relations, where relatively dense meshes of strain gauges were also employed. These types of “spot sensors”, such as strain gauges, can only provide limited information of localized values and also requires *a priori* knowledge of specific measurement locations [5]. Furthermore, applying strain gauges directly to reinforcing bars can affect the bond behavior as well as alter the characteristic behavior of the steel [6]. Other spot sensors have been used more recently to monitor the response of steel rebars in RC members, not in terms of local strains, but of local displacements. The latter can be important to measure phenomena where the relative displacement or slip between adjacent rebars is of interest, e.g., to characterize lap-splice response [7], or when absolute displacements with respect to a fixed reference are required, e.g., to determine fixed-end rotations due to strain penetration of reinforcement into the adjoining member [8]. The two latter works [7,8] used three-dimensional position monitoring systems based on a target marker instrumentation technique, consisting of gluing light-emitting diodes to rebars in concrete holes left during the casting phase, which are optically triangulated during testing. This technique has led to sensible improvements of the measurement of deformation components [8] and development of advanced mechanical models [9,10]. However, similarly to strain gauges, the previous sensors will also inevitably modify the specimen response.

To overcome the aforementioned limitations with spot sensors and to achieve the measurement of the full strain profile of reinforcement embedded in concrete, fiber-optic sensors can be used. Distributed Fiber-Optic Sensors (DFOS) allow for a high-frequency measurement of strains along the entire sensor length at high resolution [11]. When the fibers, acting as sensors, are subjected to an extension or contraction (i.e., mechanical strain), the Rayleigh scattering profile is altered due to the change in distance between the imperfections caused in the cylindrical geometry of the fibers [3,12]. A measurement system, known as interrogator, is then able to analyze the characteristics of the Rayleigh backscattered light and provide strain data. A more complete and detailed explanation of this process can be found in the literature [13,14]. Some of the unique features of DFOS include, but are not limited to, its high accuracy (e.g., ±25 με for a pitch of 0.65 mm), repeatability, stability, resistance to electromagnetic interference, protection from corrosion, low weight, small size (e.g., diameter typically less than 200 μm [3]), and low cost [12,15,16]. It is worth noting that, while the cost of the individual sensing fibers is low, at approximately US $100 per 20 m of polyimide fiber, the cost of the measurement system is still currently high [12] (e.g., 4-channel ODiSI 6104 Series system priced at approximately US $100 k).

The application of DFOS in RC structures is still in its infancy [13], despite some recent research using DFOS embedded in RC structures to bypass the difficulty of conventional instrumentation in monitoring salient internal mechanisms. For example, DFOS was attached to steel reinforcement in RC beams to study the bond performance in the inelastic range [12]. A recent study investigated the application of DFOS for various measurement scenarios in different structures, such as concrete cubes, RC slabs, and RC beams [11]. Another study investigated the strain profiles of beam-column connections by embedding four optical fibers with three different coatings at different locations [13]. More recently, six RC members were tested under uni-axial tension to investigate different types of fiber-optic cables embedded in concrete and steel reinforcement [17]. There have also been a small range of experimental investigations comparing the concrete or reinforcing steel strains derived from conventional instrumentation to that measured using the DFOS system for different structures: Brault & Hoult [18] found that the strains measured with a nylon-coated fiber compared well to a limited number of strain gauges installed on reinforcing steel embedded in RC beams; Mata-Falcón et al. [6] investigated the reliability of digital image correlation (DIC) techniques and DFOS measurements for a range of structures, including a concrete panel, and pull-out test of a reinforcing bar; Berrocal et al. [3] showed that the calculated crack positions and widths from DIC and DFOS strain measurements compared reasonably well for a RC beam governed by flexure; and Zhang et al. [13] determined that the DFOS provided strain measurements with much higher accuracy in comparison to the localized DIC strain measurements of a RC beam-column joint. While some of the aforementioned works have compared DIC techniques with DFOS measurements, there appears to be a lack of guidance or consensus in the literature as to what methods should be used to determine average and more reasonable strain values from high-resolution DIC data for large-scale tests of RC structures. To the authors’ knowledge, only one experimental investigation used DFOS in RC walls. In the study by Woods et al. [5], DFOS was bonded to the surface of the outermost layer of carbon-fiber-reinforced polymer to measure the strain distribution over the entire face of the strengthened RC wall. There has yet to be an experimental investigation using DFOS bonded to the longitudinal reinforcing steel in RC walls.

In this study, state-of-the-art technology based on the DFOS system was used to capture the strain profiles along the height of several longitudinal reinforcement bars in three large-scale RC wall units. In comparison to the aforementioned studies that have investigated the performance of DFOS against other instruments for structures with rather simple loading types, the study herein investigates the performance of this state-of-the-method instrumentation embedded in non-planar walls subjected to complex types of loading and protocols (namely axial-flexure, axial-torsion, and axial-flexure-torsion). The authors have previously published a data paper [19] which gave an overview of the experimental test setup and provided justification for the primary research aims of the experimental investigation. The large volumes of data from all instrumentation and from two of the three wall units are readily available on the publicly accessible platform *Dataverse* [20]. In the current paper, the authors present new experimental results using the available dataset that have not previously been analyzed with the aim of comparing vertical displacement and strain measurements from DFOS in the rebars within the confined core—that is, the region delimited by the confinement reinforcement—of a segment of the wall to concrete-cover surface measurements from different conventional instrumentation. The conventional measurements, including those from micrometers, linear variable differential transformers (LVDTs), and DIC techniques, were used to measure the surface deformation of the concrete at selected points or wall surfaces. The DFOS were interrogated with the ODiSI 6104 Series sensing platform (Luna Innovation Inc). Four optical fibers were attached to a total of eight longitudinal reinforcing bars (i.e., 6 × 2.86 m long Φ12 and 2 × 2.86 m long Φ8 rebars) in the west flange of each wall unit (Figure 1a) to investigate the strain profile in the cross-sectional core. Each of the three wall units was subjected to a different reverse-cyclic, quasi-static loading protocol—in-plane bending (i.e., flexure), torsion, and a combination of flexure and torsion, respectively. A constant axial load was also applied to all units. It is worth mentioning that the authors have previously published a very limited amount of some preliminary results in conference proceedings (e.g., [21,22]), and the current work provides further context and a full scope of the results, as well as other comparisons with other instrumentation. Further, the instrumentation data from the third wall unit (denoted UW3) is provided in this paper, as the aim is to compare the core and surface strain and displacement measurements using a range of different instrumentation. However, due to major construction deficiencies, a summary and overview of the overall seismic performance of this third wall unit, in comparison to other two wall units, was not originally reported in the data paper [19].

A summary of the experimental program is provided in the next section, including the test setup and instrumental layout. The DFOS measurements in the core of the wall are then compared to measurements made on the surface using conventional instruments.

## 2. Summary of Experimental Program

### 2.1. RC U-Shaped Wall Units

Three half-scale RC U-shaped walls were tested as part of an experimental program focusing on flexural and torsional seismic performance. These wall units, denoted UW1, UW2, and UW3, were designed with identical geometry and reinforcement detailing (Figure 1a), but subjected to different actions: flexure, torsion, and a combination of flexure and torsion, respectively. The wall thickness (*t_w_*), flange length (*L_f_*), and web length (*L_w_*) of all units was 100 mm, 1050 mm, and 1300 mm, respectively. These notations of the wall segments, ‘web’ and ‘flange’, correspond to that in Figure 1a, and are synonymous with the segments of other U-shaped wall tests [23,24,25,26].

The different testing positions of the walls are depicted in Figure 2a. The first wall, UW1, was subjected to in-plane bending about its minor axis, loaded to positions D and C. The second specimen, UW2, was subjected to a reverse-cyclic twist about its vertical axis, loaded to positions O+ (i.e., clockwise twist in Figure 2a) and O− (i.e., counterclockwise twist in Figure 2a). The third specimen, UW3, was subjected to a combination of in-plane bending and twisting, loaded to position D− (i.e., position D with counterclockwise twist, red cross-section in Figure 2a) and C+ (i.e., position C with clockwise twist, blue cross-section in Figure 2a). All wall specimens were subjected to a pre-compression axial load ratio of 5%, which was held constant throughout testing.

The local measurements that are presented in this paper are given at different wall loading magnitudes, such as drifts (UW1), rotations (UW2), or both (UW3). To clarify, the drift (*δ*) of the wall is the ratio of the in-plane translational displacement to the height of application of the lateral displacement Δ (i.e., *δ* = Δ/*h_s_* = Δ/2250). The applied rotation (*θ*) to the wall top collar was calculated by dividing the translational displacements of the west and east flanges (Δ*_WF_* and Δ*_EF_*, respectively) by the centerline distance between the flanges (i.e., θ = (Δ*_WF_* − Δ*_EF_*)/1.2 in millirad).

More information regarding the specifics of the tests, including the material properties of the concrete and steel, the test setup, loading protocols, and summary of the failure modes observed, can be found in the data paper [19].

### 2.2. Test Instrumentation

A 3-dimensional schematic illustration of the primary sources of instrumentation used to record data during the test is given in Figure 2b. The instrumentation depicted in Figure 2b includes Distributed Fiber-Optic Sensors (DFOS), linear variable differential transformers (LVDTs), a speckle pattern to capture the surfaces’ deformations using Digital Image Correlation (DIC), and micrometers. The specifics of these instruments are given in the following sub-sections.

#### 2.2.1. Distributed Fiber-Optic Strain Sensing

Each wall unit had DFOS bonded to a total of eight longitudinal rebars located in the west flange as indicated by the red dots in Figure 1a. DFOS were implemented using the LUNA optical distributed sensor interrogator 4-channel ODiSI 6104 Series system. Polyimide-coated single-mode sensing fiber was employed. Although nylon-coated fiber-optical cables have been shown, in one prior investigation, to produce more reasonable strain results in RC beams [18], the polyimide-coated fibers that were used here were chemically bonded to the fiber-optic core [11] and do not exhibit slip at the locations of sudden strain change [27], which can result in more accurate strain measurements in the areas of localized deterioration [12]. For all fiber-optic cables, the spatial resolution was set to 0.65 mm with a sampling rate of 3.125 Hz. This small gauge length of 0.65 mm and the sampling frequency used to measure and record the strains from the DFOS potentially influenced the signal-to-noise ratio in the recorded data [3]. Thus, as practiced in other research, the DFOS strain data presented herein has been filtered using a moving average over a length of approximately 10 mm to reduce noise levels [3,28]. The testing of the three U-shaped wall units was performed at the LEMSC laboratory, which has a heating, ventilation, and air-conditioning (i.e., HVAC) system that helps to regulate the temperature and humidity. Since it is unlikely that there were any significant thermal-dependent changes to the corresponding strain measurements acquired from these quasi-static tests, the following investigation assumes that there were no thermal effects on the fiber optic measurements.

As only four channels were available with the interrogator, one fiber-optic cable was utilized to measure the strains of two longitudinal rebars simultaneously. As such, the two layers of the longitudinal reinforcement across the thickness of the wall were welded together in the foundation and in the head. Figure 3 illustrates the structural drawings of the longitudinal rebars bonded with DFOS, which are shown to have a rounded segment welded in the collar (head) of the wall, whereas a straight steel bar segment is used to join the two rebars at the base.

A small groove (i.e., slit of 1 mm width and depth) was cut along the rebar (Figure 4b), which has been widely practiced in previous research for attaching the DFOS to steel reinforcement and helping to prevent a premature failure of the fiber [11,17,29,30,31]. Ideally, the groove is milled along the longitudinal rib of the rebar as to not affect the cross-section or the bond from the transverse ribs. However, this could not be practiced in the experiments herein due to laboratory equipment limitations, and instead, the groove was milled along the rebar as shown in Figure 4b. After some initial trials, a general-purpose adhesive (“Loctite 401”, cyanoacrylate technology, or “CYN” in Figure 4a) was used to bond the fiber into the groove of the bar, which has also been successfully used in the literature for this purpose [3,11,18,32]. Some researchers have concluded that applying a thin layer of protection between the optical fiber and the concrete can achieve more reasonable strain profiles and, more importantly, help to achieve measuring strains after concrete cracking has occurred [18,33]. Thus, a silicone-based epoxy (“SI” in Figure 4a) was also used to provide a coating of protection between the glued-fiber and the concrete. Figure 4c shows an orange plastic cone that was used to house the end of the rebar. The cone was attached to the rebar using heat-shrink tubing to protect the connection point and termination (i.e., the strain relief region) of the fiber during the preparation of the steel cage and pouring of concrete.

There currently appears to be a limitation in using DFOS for reinforcing steel that is expected to behave inelastically where fiber-optical sensing is suggested to be not suitable for strains much greater than yield. For example, polyimide-coated fibers have achieved measurable strains (*ε*) up to 1% in previous experiments [11]. Others have suggested an even lower strain limitation where the DFOS measurements were unreliable as soon as yielding (*ε* ≈ 0.25–0.29%) of the steel reinforcement commenced [18]. The technical datasheet for the ODiSI 6104 interrogator system used in this research states to have a strain measurement range of ±1.5%. However, in reviewing the literature on this topic, the strain measurement range appears to be largely dependent on the expected strain gradients. High inelastic strain gradients have previously been shown to result in a significant deterioration in data quality of reinforcing steel embedded in concrete beam-column joints [13]. This is because the interrogator unit, similar to that used in this research, was unable to analyze and detect peaks in the frequency spectra. The same observation has been stated in Mata-Falcón et al. [6], where errors occurred as soon as the steel reached its yield point. Improvements in the interrogator unit and corresponding software may in the future overcome this limitation, since the glass fiber itself ruptures at a much higher strain of approximately 4% [6]. It is theoretically possible to recapture new reference data sets containing the unique Rayleigh scatter pattern of a single fiber sensor when it reaches large strains (e.g., 1%) to take it to a larger strain range. This operation is known as rekeying, and re-centers the strain around a new point and removes strain gradients. However, in practice, even for slow monotonic loading rates—such as the quasi-static loading protocols used to test the wall units in this research—establishing new “key” measurements, thereby resetting the maximum possible strain measurement range, is very difficult and bordering on the impossible [6]. To the authors’ knowledge, only two research investigations have reported accurate readings of large inelastic reinforcement strains: Malek et al. [12] likely used the “rekeying” method to measure strains up to approximately 2.5% using DFOS embedded in scaled RC beams. Poldon et al. [34] used a similar method, which they referred to as the “leapfrog technique”, to measure the inelastic strain profiles, with strains up to approximately 1.6%, using DFOS in a simply supported RC beam governed by flexure. Importantly, the loading applied to the beams in the former of these two research investigations was monotonic, allowing the DFOS system to be “rekeyed”, for example, when strain limits were reached, and the cumulative sum of the inelastic strains could be calculated in post-processing. Given that multi-channel strain sensing was used in the research program herein, coupled with the reverse-cyclic loading protocols, the method to “rekey” was deemed to be too difficult. The authors acknowledge that it is possible that other sensor systems may be more applicable in potentially measuring larger strain values, such as that based on Brillouin scattering [35,36], rather than the Rayleigh-based distributed sensors used here. However, to the knowledge of the authors, Brillouin sensors, which are often used in structural health monitoring [37,38], generally provide lower resolutions (e.g., 10–15 cm [36] or 1.5 m [35]), which would present another limiting factor in the strain measurements for the experimental tests on the scaled wall units presented herein. Addressing the aforementioned practical difficulties would be a welcome future development for the application of this Rayleigh-based sensing technology to RC structural experimental testing. In summary, it is expected that the present study is limited to examine DFOS strains up to, and only just beyond, first yielding of each of the respective reinforcing bars.

#### 2.2.2. Linear Variable Differential Transformers (LVDTs)

Three LVDT arrays were attached to each of the wall units to measure the average axial strain of the concrete surface along the corresponding base length of each LVDT. The LVDT arrays were located on the surface of the two boundary ends of the flanges, depicted in Figure 2b, and on the west flange towards the web (denoted here as the west flange-web intersection). For the research undertaken and presented in this paper, only the two chains of LVDTs located on the west flange (on the intersection with the web and on the boundary end) will be used, since they correspond to wall regions that are monitored simultaneously by DFOS. The locations of the LVDT arrays to the north (i.e., flange-web intersection) and south (i.e., boundary end) and the corresponding base lengths of each LVDT are illustrated in Figure 3a. It is worth noting that two LVDTs were used within the wall thickness at the base of the boundary end (to the south), one interior and another exterior (relative to the center of the wall) with base lengths of 600 mm and 450 mm, respectively (Figure 3b).

#### 2.2.3. Digital Image Correlation (DIC)

A speckle pattern for digital image correlation (DIC) measurements was applied on two outside surfaces of the wall units. For this research, only some of the DIC results for the web region of the wall, depicted in Figure 5a, will be used. The speckle pattern covered the full height of the wall units from the base to its height of 2 m. The speckle pattern was applied by a stencil or stamp roller to produce a random pattern of black dots with an approximate diameter of 2.5 mm. Two three-dimensional DIC systems were installed, where 2 sets of two high resolution (12 megapixel) cameras recorded monochrome (i.e., black and white) images at a frequency of 0.2 Hz (i.e., every 5 s) during testing. This frequency was decreased to 0.1 Hz and 0.05 Hz for later loading stages that required a longer testing time to achieve large displacements and rotations. The images were processed using Istra 4D [39], where the strain field was exported into tab-separated values to be further processed and analyzed in MATLAB [40]. An example of the strain output in the form of a heatmap is shown in Figure 5a (strain values not indicated). For this research investigation, only the strains captured by the 100 mm × 2000 mm surface, depicted in Figure 5b, corresponding to the web boundary towards the west will be used. The northwestern wall corner “column” of 100 mm × 100 mm × 2000 mm (Figure 5b) provides an ideal opportunity to compare the LVDT chain and DIC surface measurements of the web to the internal strain measurements determined by the DFOS bonded to the two rebars, as is also depicted in Figure 5b.

There was some difficulty in achieving DIC measurements of the web for each wall test due to some geometrical constraints in the laboratory, which affected the placement and angle of the cameras. While high-resolution data was achieved for UW1 and UW3, a lower resolution of information was achieved for UW2. The corresponding correlation parameters used to process the DIC files for each wall unit is given in Table 1. The different resolutions achieved for each wall unit can be reflected by the grid spacing of the DIC data in Table 1. Furthermore, some information was not captured by the DIC system for the web of some units: strain measurements above a height of 1600 mm and 1900 mm were not captured for UW2 and UW3, respectively. While these limitations are far from ideal, the strain measurements still provide a broadly sufficient opportunity for comparisons to other instrumentation, as will be analyzed in Section 3.2.

#### 2.2.4. Micrometers

A total of 16 digital Mitutoyo micrometers were used to measure the vertical displacement of the concrete wall surface along the interior perimeter at the base. The micrometers, labelled M1 through to M16 in Figure 1a, were strategically placed at the locations corresponding to the approximate placement of some of the longitudinal reinforcing steel bars. The vertical displacements were measured over a distance of 60 mm from the base, as shown in Figure 6. Two different types of micrometers were used: the round Digimatic indicators (M3, M4, M7, M8, M9, M10, M13, and M14 in Figure 1a) have a range of 12.7 mm and were placed more centrally, where vertical elongation was expected to remain lower; the other micrometers (M1, M2, M5, M6, M11, M12, M15, and M16) had a greater range of 25.4 mm and were placed in the boundary regions, where vertical elongation was expected to be higher. The placement of these micrometers provides an opportunity to compare their measured vertical displacement, captured on the inside surface of the wall, to that calculated from the DFOS bonded to the rebars (i.e., M1, M2, and M5 in Figure 1a).

## 3. Surface and Core Results

In this section, experimental strain and displacement results will be presented, derived from core instrumentation, and compared to surface instrumentation.

### 3.1. LVDTs and DFOS

This section compares the average strains calculated from the two arrays of LVDTs, located on the west flange towards the north (i.e., flange-web intersection) and south (boundary end), with the channels 1 (“ch1”) and 4 (“ch4”) of the DFOS system (Figure 3), respectively.

Figure 7 presents the strain profiles derived from DFOS and LVDTs for specimen UW1, which was subjected to in-plane bending about its minor axis (positions C and D, Figure 2a) at different imposed drifts. Both of the LVDTs at the base of the boundary end, i.e., the exterior and interior LVDTs pictured in Figure 6, with base lengths of 450 mm and 600 mm, respectively, were used to derive the base strains in Figure 7b. As expected, the base strain for the 450 mm LVDT is larger than for the 600 mm LVDT, since the contribution of the strain penetration in the foundation is unsurprisingly more significant than the wall deformation between the heights of 450 mm and 600 mm. As is well known in the structural and seismic design and analysis of RC structures, strain penetration effects can contribute to a significant amount of the overall lateral displacement capacity [41]. Additionally, it can be observed that the LVDT tensile strains at the base are, in general, greater than those measured on average by the DFOS. For example, in Figure 7b, when the wall is pushed towards position D and the west flange boundary end is in tension, the base strain determined from the exterior bottom LVDT (450 mm base length) is approximately 6 mm/m for a drift (*δ*) of 0.6%, whereas ch4 of the DFOS measures an average strain of 3.6 mm/m (Figure 7a) over the same base length of 450 mm. It is interesting to note that the DFOS on the exterior rebar outputs a strain at the foundation wall interface crack of 4.6 mm/m, attaining a local minimum of 3.3 mm/m at mid-crack spacing above (i.e., at a height of approximately 40 mm); such variation will increase for walls UW2 and UW3, as it will be shown. In reality, recalling that the DFOS results correspond to a moving average over a length of approximately 10 mm, the peak strain of 4.6 mm/m hides, in fact, a larger local strain of 6.1 mm/m. The discrepancy between base LVDT measurements and DFOS can again be explained due to the effects of strain penetration, which are not considered in the LVDT-derived strain outputs. Had strain penetration effects been considered, they would increase the required effective base length for strain computations. In fact, without the strains from the DFOS, which clearly capture the strain penetration into the foundation, it is very difficult to determine the contribution to the LVDT strain output coming from deformations inside the foundation. Other investigations have proposed methods aiming at approximately removing deformation contributions in RC members from outside the gauged region [42]. Please also refer to Section 3.3. LVDT chains, such as those used here at the extreme compression and tension fiber regions of the wall, are used in most wall tests to determine average curvatures along the wall height [43]. However, without consideration of strain penetration effects, it is inevitable that base curvatures are overestimated using these LVDT average strains. Previous researchers have recommended that the base length of the lowest instrument in the LVDT chain be sufficiently small, such that it only extends over the base crack between the wall and foundation [43]. However, even when using a sufficiently small base length, the resulting strains and curvatures from the LVDTs will not be accurately represented without defining a meaningful base length that includes the strain penetration [44]. The average strains determined by the LVDTs higher up the wall from the base appear to provide more reasonable estimates in comparison to the DFOS strain profiles in Figure 7.

The measured strain profiles for the interior and exterior longitudinal rebars (along the thickness) are almost identical, which was expected with the type of loading subjected to unit UW1. In addition to the gradual change of strain along the wall height, local strain peaks can be observed from the DFOS in Figure 7a,c, which correspond to the location of flexural cracks. Using the tensile strain profile from channel 4 of the DFOS for *δ* = 0.6% (Figure 7a, position D), the average spacing of these strain peaks (crack spacing) was calculated to be 68 mm, which is approximately the spacing of the stirrups (≈ 75 mm), corroborating previous observations in beams [13]. 

In comparison to the DFOS strain profiles, the resulting LVDT strain profiles appear to be very dependent on where the surface sensors are placed. A good example is shown in Figure 7d, where two peaks in tensile strain can be observed at a height of approximately 0.85 m and 1.35 m from the foundation. In comparison to the gradual and smooth DFOS strains in Figure 7c, these peaks of tensile strain calculated from the LVDTs are likely caused by asymmetric inclusion of concrete surface cracks in the gauged areas.

Figure 8 presents the strain profiles derived from DFOS and LVDTs for specimen UW2, which was subjected to a reverse-cyclic rotation about its vertical axis (positions O+ and O−, Figure 2a). Unfortunately, one of the LVDTs on the face of the west flange-web intersection, from a height between 900 mm and 1200 mm, malfunctioned, and hence there is a loss of information in Figure 8d. Many of the same comparisons observed for UW1 in the previous paragraphs can be made here with respect to the results in Figure 8 for UW2. However, unlike the loading imposed to UW1, the rotation applied to UW2 resulted in a strain gradient across the flange thickness, and hence a difference in the DFOS strain profiles was measured for the interior and exterior rebars, particularly for the west flange boundary end in Figure 8a. This difference in strain, which is due to the combination of circulatory and warping torsion [45] of the flange, is not captured with the array of LVDTs, although there is clearly some difference in tensile strains calculated by the two base LVDTs in Figure 8b. Thus, the LVDT surface instrumentation, in comparison to the DFOS, is unable to capture the detailed information about the behavior of the wall with regards to strain gradients across the thickness of the wall. Figure 8a,c also clearly show the existence of a steep moment gradient along the flange height, which can only be very insufficiently captured by the LVDTs. It is hence not surprising that the peak strain measured by channel 4 of the DFOS at the foundation wall interface, which takes a value of 6.5 mm/m, is larger than the average strain of 5.4 mm/m given by the bottom LVDT, unlike what happened for UW1. The strains measured by the DFOS attached to the interior bar in the boundary end of UW2 capture the compression at the head of the wall and tensile strains at the base when loaded to position O+ (Figure 8a). Such distribution is not captured by the LVDT array in the boundary end (Figure 8b), and instead, the average strains at the top of the wall, over a base length of 500 mm, are close to zero. As explained in the data paper [19], it is postulated that the application of the axial load to the walls, which was distributed evenly to three vertical actuators, is responsible for the partial warping restraint, corresponding to this observed behavior. It is worth mentioning that while it was possible to place two LVDTs at the base of the boundary end, two layers of LVDTs could not be achieved along the full height of the west flange boundary end because of the small thickness of 100 mm.

Figure 9 presents the strain profiles derived from the DFOS and LVDTs for specimen UW3, which was simultaneously subjected to both an in-plane translation and twist (positions D- and C+, Figure 2a). Some of the observations for the comparisons made between the LVDTs and DFOS strains for UW1 and UW2 are consistent with the strain profiles in Figure 9**.** for UW3. For this wall unit, a drift (δ) and simultaneous rotation (θ) of 0.8% and 8 mrad, respectively, result in DFOS tensile strains in the boundary end rebars greater than yield (>2.9 mm/m) at position D− (Figure 9a). For the exterior bar, in particular, a peak tensile strain of 12.6 mm/m was attained in the first crack above the foundation wall interface. This peak strain value contrasts with local minima of 4.4 mm/m and 3.9 mm/m at mid-crack spacing below and above, respectively (i.e., at a depth of −50 mm and height of 156 mm from the foundation level). It is likely that the very large reported values of measured tensile strains by the DFOS do not exactly correspond to actual strains inside the steel rebars. In fact, at the abovementioned level of loading, some information loss in the regions close to the localized strain peaks may occur. As suggested by Brault and Hoult [18], inaccuracy in the strain readings from the DFOS once plasticity occurs is likely to be caused by one or a combination of factors: (i) After deterioration of the bond between the concrete and the steel reinforcing, pinching of the fiber occurs, which affects the light transmitting through the fiber optic core; (ii) Significant crack openings with the potential to shear the surface of the fiber; and (iii) Large strain gradients, which result in measuring difficulties for the interrogator (i.e., analyzer) [46]. For the results herein, it is unlikely that the first two factors accounted for the inaccurate readings. As the loading imposed is reverse-cyclic, the wall returns to its centered position (i.e., position O), where the strains decrease, and the strain information returns as the optical fiber is still intact. However, a net positive strain is measured by the DFOS within these regions of the localized plasticity from previous load stages. These localized peaks in tensile strain from the DFOS continue to the next load stage, at position C+, when the flange end is in compression (Figure 9a). It is possible that the rebar in these localized areas has, on the previous load cycle, yielded in tension, deformed inelastically, and attains permanent damage that results in residual tension for the next load cycle. However, it is also possible that the reverse-cyclic tension-compression yielding of the rebar across these localized cracks has caused the fiber-optic cable to buckle, something that has been postulated to occur in previous experiments using DFOS in RC beam-column joints [13]. Another explanation for this net-tension reading by the DFOS is due to thermal effects from plastic deformation of the rebars at these localized areas, which has not been accounted for in these experiments.

### 3.2. DIC and DFOS

In this section, the DFOS core strain measurements will be compared to the DIC surface measurements. Specifically, the DFOS strains from channel 1 (“ch1”, Figure 3a) will be compared to the surface strain field of a boundary column, as Figure 5 depicts.

High-to-medium resolution strain fields from DIC typically need to be smeared using a moving average to result in values that are more representative of the rebar and concrete strains of large-scale structural elements and more compatible with most of the existing stress-strain models. For example, a base length equivalent to the wall thickness (*t_w_*) was previously used as a basis to average the high-resolution (20mm grid spacing) DIC data from the test of two RC walls [47]. Similarly, the DIC data from two tested planar RC walls used a base length of 100 mm (0.5*t_w_*) to average the strain distribution across the faces of the units [48]. However, the resulting strains from the aforementioned research investigations were not validated against other instrumentation, such as LVDTs. In another study focusing on RC beams [3], a procedure was used to average DIC surface strains to calculate crack widths in comparison to the estimates from the DFOS strains, which involved applying a threshold strain limit to remove noise, fitting a Gaussian distribution, and taking a mean. However, this resulted in “corrected” DIC strain profiles with distinctive peaks that were larger than the “smoothed” strain profile obtained from the DFOS, as the focus of the study was on providing accurate crack width calculations and not on strain profiles. This is explained by Ruocci et al. [49], where the crack is locally detected by the DIC as a displacement discontinuity leading to a peak in the strain fields. The local strain peaks from DIC were also visually interpreted in an experimental study on a beam-column joint, resulting in the authors to conclude that the DFOS provided more reliable strain measurements [13]. In fact, the majority of literature reporting on RC wall tests and using DIC focus on crack detection [50,51] or measuring crack width and spacing [49], rather than accurate strain profile measurements. As such, and as suggested in the introduction, there appears to be a lack of guidance as to what sort of averaging method or procedure should be used on high-resolution DIC strain data to provide more accurate strain measurements in the testing of large-scale RC structures.

For the above reasons, in the initial part of this section, the authors use DIC and DFOS results from one wall specimen to recommend a base length value. Figure 10 presents the longitudinal DIC strain values of UW1’s boundary column of the flange-web intersection in tension and compression (position C and position D, respectively, for a drift δ of 0.6%). The average of the longitudinal strains determined across the 100 mm width of the boundary column (Figure 5b) was computed, whereas different base lengths (*B_l_*) were used to take a moving average vertically: (i) No moving average (“uncorrected”); (ii) *B_l_* = 50 mm; (iii) *B_l_* = 100 mm; and (iv) *B_l_* = 200 mm. The latter of these values represents an upper limit, equivalent to 2.0 × the thickness of the wall (2.0*t_w_*). The 50 mm and 100 mm were chosen based on 0.5 × and 1.0 × the thickness *t_w_*. For reference, the DFOS strains (average of the two rebars, black solid line) and calculated strains from the LVDT array (black dashed line) are superimposed in these figures. As mentioned previously in Section 3.1, the loading applied to UW1 resulted in closely resembling DFOS strain profiles between the two layers of rebars, which justified taking the average to result in a single profile here. The “uncorrected” DIC strains, with no applied moving average, show peaks of large tensile strains (Figure 10b), which correspond to the local strains captured across small, open flexural cracks at the surface of the concrete. The average of the spacing between the tensile peaks (or crack spacing) is 64.7 mm, which is close to the designed spacing of the transverse confinement (= 75 mm). While many factors ultimately determine the crack spacing of a RC member in tension [9], the spacing of the transverse reinforcement is a primary factor. Interestingly, the average spacing distance between the “uncorrected” DIC compression strain peaks (Figure 10a) is 31.9 mm, approximately half of the spacing distance between the tensile peaks. In this case, it is likely that compression strain peaks from the DIC coincide with the location of cracks closing, as well as the concrete in between the cracks, whereas the reinforcing bar in compression exhibits a smooth strain profile (as indicated with the DFOS strains). If it is assumed that the reference strains from the DFOS and LVDTs are more reliable indicators of the strain profile, the DIC tensile strains with *B_l_* = 200 mm appear to provide better estimates representative of the compression and tensile strain profiles. There is a clear difference of DFOS measured strains and the DIC calculated strains at the base of the wall in both compression and tension (Figure 10a and Figure 10b, respectively). Firstly, when the boundary column under consideration is in tension (Position C, Figure 10b), the tension at the base is not adequately captured by the DIC technique due to strain penetration not being considered [47]. The uncorrected DIC strains at the very base of this column have been ignored for the purposes of deriving the corrected DIC strain profiles due to the unrealistically high concreated strain values, which, in some cases, will result in a poor estimate of the base tensile strain. Secondly, when the boundary column is in compression (Position D, Figure 10a), it is expected, under the current flexural loading, that compressive strains in the concrete surface, captured by DIC, would be greater than the compression in the steel bars, captured by the DFOS, which are embedded at some 25 mm from the surface of the boundary column. For large drift levels towards Position D, the neutral axis is located somewhere in the web of the wall (see Figure 1a for the wall segment definitions). Therefore, the web surface of the wall captured by the DIC, or in this case, the surface of boundary column under consideration, represents the extreme compression fiber region (for loading towards position D), which is expected to experience greater compression strains than the rebars closer to the neutral axis.

Figure 11a–d and Figure 11e–h present the DIC tensile and compression strain profiles, respectively, from the surface of the boundary column for unit UW1, subjected to in-plane bending, for four different levels of drift (*δ*): 0.2%, 0.4%, 0.6%, and 0.8%. Drift levels greater than this were found to cause some information loss in channel 1 of the DFOS. A heatmap is presented to the left side of each plot, representing the uncorrected, raw DIC strain data (in units of mm/m) for the entire 100 × 2000 mm^2^ surface of the boundary column (illustrated in Figure 5b). The right side of each figure plots the uncorrected DIC strains (blue thin lines), where the average of the longitudinal DIC strains across the 100 mm width of the boundary column have been computed for each vertical increment. Superimposed in these plots are the corrected (vertical moving average with *B_l_* = 200 mm) DIC strains (red thick lines) and the strain profiles measured from the DFOS bonded to the two *Φ*12 mm rebars (black solid lines). Note again that for the loading applied to UW1 (i.e., in-plane loading to position C and D, Figure 2a), the resulting DFOS strain profiles of these two rebars were essentially the same. Overall, the corrected DIC strains provide more representative estimates for these drift levels, assuming that the reference strains from the DFOS are more reliable indicators of the strain profile. Due to the fact that the DIC technique captures the tensile strains across the cracks of the concrete surface, slightly larger tensile strain oscillations can be observed in comparison to the smoother DFOS strains occurring in the section core. This smoothness is due to the bond stress transfer that occurs between the steel rebar and the surrounding cover concrete (recall that the fiber-optic cables are placed on the outside of the steel rebar, to the east and to the west for the exterior and interior bar, respectively—see Figure 4a,b), of which only the surface manifestation of this phenomenon is captured by the DIC. Several other factors could have contributed to the limited discrepancies of strain profiles observed between the surface and core measurements, with the most obvious being the difference in the location of the surface plane (i.e., the web of the wall, Figure 5) captured by the DIC and the location of the embedded rebars with DFOS in the boundary column, some 25 mm from the web surface. Furthermore, local bending of steel reinforcement has been shown to produce variations in the magnitude of the resulting strain measurements [52], where, for this study, the fiber-optical cables are located on the outside of the rebar. Other factors include the strain “transfer effect”, which could result in the strains determined by the sensors being different from the strains of the host structure (i.e., the steel rebar) [15,53,54]. While the authors took a number of precautions to reduce these strain transfer effects [55] (e.g., see Section 2.2.1, consideration of adhesive type, engraving a small groove, fiber bonded directly to the reinforcement [3], etc.), these effects cannot be completely ruled out.

For the boundary column of unit UW2, which was subjected to a reverse-cyclic torsional rotation, Figure 12a–d and Figure 12e–h present the DIC tensile and compression strain profiles, respectively, for four different rotation (*θ*) levels: 15 mrad, 20 mrad, 25 mrad, and 30 mrad. This type of loading caused slight differences in the DFOS strain profiles of the two longitudinal rebars, represented by the black thick lines. While the rather gradual gradient of the strain profiles measured from channel 1 of the DFOS system for UW2 meant that strains greater than yield could be achieved, rotation levels greater than 30 mrad started to cause some information loss in this channel. The latter can be observed at the base of the boundary column in Figure 12c,d, where loss of DFOS data occurs for strains greater than 10–15 mm/m, corresponding to the limits given in the technical dataset for the LUNA system used (see Section 2.2.1). Note that the DFOS in both layers of the rebars measured the localized strain increase at the same location. Interestingly, the uncorrected (raw) DIC strain peak towards the base in Figure 12c and Figure 12d compare reasonably well to the strain peak measured with the DFOS, corresponding to where the information loss occurs. These strain levels also cause a noticeable net-tension strain measurement in the DFOS at the next load stage in compression, presented in Figure 12h with the black thick lines. Note that the DFOS in both layers of the rebars measured this localized net-tension in strain measurement. A list of possible explanations for this was included in Section 3.1, with a possibility of this reflecting a real, localized phenomenon. However, the maximum DIC compression strains appear to concentrate at approximately 100 mm from the base in Figure 12h, which is the same location of the maximum tensile strains in the previous load stage in Figure 12d. Thus, it is also possible, and maybe more likely, that this level of rotation (θ = −30 mrad) has caused the DFOS to be pinched or buckle, resulting in a false reading of strain at this location. This explanation appears to be consistent with the reasons given in Zhang et al. [13], where the phenomenon was observed for DFOS bonded to rebar embedded in a RC beam-column joint that was reverse-cyclically tested. Overall, the corrected DIC data in Figure 12a–d (red thick lines) appear to reasonably represent the strain profiles for all levels of rotation considered in comparison to the tensile strain profiles measured by the DFOS. These results also show that the recommended correction, using a simple moving average with a base length *B_l_* of 200 mm, can be applied to DIC data of high, medium, and low resolution, with the latter of these represented by the DIC data attained for unit UW2. For example, while a clear smearing effect of the uncorrected DIC data could be observed in Figure 11 for UW1, the effects are less prominent in Figure 12 for UW2, particularly for the compression strain profiles in Figure 12e–h. Thus, it appears that the recommended correction method for DIC strains has its greatest effect when the resulting vertical grid spacing is much less than the base length *B_l_* recommended here of 200 mm. However, the corrected DIC compression strain profiles in Figure 12e–h appear to overestimate the compression strain profiles of the rebars measured by the DFOS. Similar to the DIC and DFOS results for UW1 in Figure 11e–h, the large compression strains could be expected, given that the DIC is measuring the compression strains of the concrete surface, representing the extreme compression fiber region; instead, the DFOS are attached to the longitudinal rebar embedded at some cover distance away from the surface.

The DIC data attained from the web of UW3 had the highest resolutions of all three wall units with a resulting vertical grid spacing of 5 mm (Figure 2a). This is compared to a vertical grid spacing of 10 mm and 20 mm for UW1 and UW2, respectively. Wall unit UW3 was subjected to both a reverse-cyclic torsional rotation and a translational push-pull, simultaneously (i.e., position C+ and D− in Figure 2a). The uncorrected and corrected DIC tensile and compressive strains for the web boundary column of UW3 are presented in Figure 13a–d and Figure 13e–h, respectively, for four different drift-rotation (*δ*-*θ*) levels: *δ* = 0.1% and *θ* = 1 mrad, *δ* = 0.2% and *θ* = 2 mrad, *δ* = 0.3% and *θ* = 3 mrad, and *δ* = 0.4% and *θ* = 4 mrad. The high-resolution, uncorrected DIC data in Figure 13 (blue thin lines) results in high-frequency peaks of tensile and compressive strain, which make it difficult to interpret without processing the data using, for example, a moving average (red thick lines). In comparison to the DFOS strain profiles in Figure 13 (black dashed lines), the corrected DIC strain profiles compare reasonably well. Larger DIC tensile strains can be observed at the base of the boundary column (i.e., Figure 13d), which is a consequence of this surface instrumentation method not being able to measure the strain penetration into the foundation, similar to the observations for the LVDTs in Section 3.1. There is also some small, but noticeable, positive (tensile) surface strains in Figure 13h from above mid-height, calculated from the corrected (red thick line) DIC data, which is not measured by the DFOS (black thick lines). It is worth noting that this type of strain gradient was similarly illustrated in the data paper [19] and measured by the DFOS for UW2 (i.e., pure rotation), particularly for the rebars of channel 4 in the west flange boundary end.

### 3.3. Micrometers and DFOS

In this section, the vertical displacement as measured from three of the sixteen Mitutoyo micrometers, specifically M5, M2, and M1 in Figure 1a, are compared to the displacement from the DFOS, specifically channels (“ch”) 2, 3, and 4 in Figure 3a. The vertical displacement (Δ*_v_*) from the DFOS is calculated by integrating the measured strain over the length (*L*) of the undeformed bar (and optical fiber) defined below, according to Equation (1).
(1)$$\Delta_{v}=\int_{0}^{L}\varepsilon dz$$
where *ε* is the vertical strain measured from the DFOS.

The length *L* considered here for calculating Δ*_v_* is approximately 0.34 m, which is the anchorage length of the rebar into the foundation. However, for each rebar and wall unit, the specific depth into the foundation is more accurately determined from pre-established gauge locations of the DFOS along the rebar. For simplicity, this displacement is called “anchorage slip”. Calculating the anchorage slip (Δ*_v_*, Equation (1)) using this depth of approximately 0.34 m provides a lower bound estimate, as the micrometer surface sensors were attached to the wall at approximately 60 mm above the foundation. Furthermore, as the surface sensors (micrometers) were placed on the inside of the wall (Figure 14), only the DFOS bonded to the interior rebar is used for comparison purposes here.

Figure 14 plots the resulting Δ*_v_* for UW1 as calculated with the DFOS strains (black solid line) and as measured from the micrometers (grey solid line), where the difference between the two is also indicated in the figure (red dashed line). Load stage (LS) numbers 5 and 18 were not recorded by the DFOS system due to instrumentation errors, and hence, the data is not present in Figure 14 for these two LS. It is worth noting that for UW1, the measurement point of the three micrometers was found to be below the bottom-most flexural wall crack; this crack should be distinguished from the base crack at the foundation wall region. Overall, the anchorage slip calculated from the DFOS from all three channels and for the LS range considered here reasonably matches the measurements from the three micrometers, albeit there are some small differences for some load stages. As wall unit UW1 was loaded in-plane parallel to the flanges, strain gradients across the thickness of the two wall flanges (see Figure 1a for segment definition) are expected to be unimportant (at least until out-of-plane deformations are developed during the last load stages), and are mainly attributable to inevitable small construction and loading asymmetries. In fact, the DFOS strain profiles from the two rebars across the thickness show negligible variations. Thus, the small differences in Δ*_v_* that are observed in Figure 14 between each pair of instruments are instead likely to be due to the slip of the rebar from the concrete, where the micrometers are only able to measure a fraction of the vertical deformation at the surface.

Wall unit UW2 was loaded in reverse-cyclic rotation, and, due to a combination of warping and circulatory torsion, a strain gradient was observed through the thickness of the flanges with the DFOS strain profiles. It was therefore interesting to compare the anchorage slip as measured by the micrometers and calculated with the DFOS strains. The results are shown in Figure 15, and they are largely consistent with those observed for UW1, subjected to in-plane flexure. However, channel 4 (i.e., “Ch4” in Figure 15c) of the DFOS, located at the outermost region of the flange boundary end, provides reasonable estimates in tension (i.e., positive Δ*_v_* values), but lower estimates in compression compared to the micrometer measurements. This observation could be a product of the lower bond depth used to calculated the DFOS Δ*_v_* as well as the strain gradient across the thickness of the wall, where larger tensile and compressive strains are expected towards the inside of the wall. In fact, as shown in Figure 6b, the offset distance from the attachment of the metal plate to the wall and the application of the tip of the micrometer will amplify the anchorage slip, as measured by the micrometer, due to the strain gradient across the thickness of the wall. Channel 3 of the DFOS (Figure 15b), which was bonded to rebars also located within the boundary end of the west flange, was found to produce higher values of the anchorage slip throughout the LS range in comparison to the measurements for M2, but only in tension. In compression, micrometer M2 measured larger values of displacement compared to the computed Δ*_v_* from Ch3. Regarding the west flange-web intersection (see Figure 1a), channel 2 of the DFOS is shown to compare reasonably well with micrometer M5 in Figure 15a. Within this latter comparison, it is noted that there are at least two LS levels that produce some discrepancies between the different sensors: at LS 12–13 and LS 21–22, when the wall is rotated to position O+ (Figure 2a), Figure 15a shows micrometer M5 measuring a positive (tensile) Δ*_v_*, whereas the value calculated from Ch2 of the DFOS is insignificant and close to zero. A possible explanation is that, at this wall position (O+), the warping of the wall is likely to cause a distribution of longitudinal strains across the wall flange at the base, but with a diagonal neutral axis (across the thickness) close to the position of this rebar and micrometer (Ch2 and M5 in Figure 15a). On further investigation, it was found that, at these two LS levels, Ch2 of the DFOS indicate that a large portion of the development length of the interior rebar into the foundation was in tension, contrasting the exterior rebar, which measured compression strains throughout its profile. This further suggests a complicated strain gradient through the thickness, and into the foundation, due to the loading imposed with the neutral axis, at these LS levels, within the vicinity of these two rebars. In fact, the two following LS levels that rotate the wall to position O- (i.e., LS 14–15 and LS22–23) also show the micrometer return to a measured Δ*_v_* of close to zero before increasing in positive (tension) anchorage slip again, further substantiating the diagonal neutral axis theory.

Test unit UW3 was subjected to a combination of reverse-cyclic torsional rotation and translation simultaneously, corresponding to positions D− and C+ (Figure 2a). The corresponding Δ*_v_* measurements from the micrometers of UW3 are presented in Figure 16, which can be compared to the calculated anchorage slip using the DFOS strains. The comparisons between the core and surface sensors look reasonable up to load stage 13. At the end of LS 13 (Position D-), an increase in the measured Δ*_v_* by M1 and M2 (Figure 16c and Figure 16b, respectively) can be observed, with respect to that computed from the DFOS. For each LS that follows corresponding to a wall push to Position D− (i.e., LS 16-17, 19-20, and 22-23 in Figure 16), a significant increase in the measured anchorage slip can be observed from the micrometers M1 and M2 in comparison to the computed Δ*_v_* from channels 3 and 4 of the DFOS. One explanation for this increase of the anchorage slip measured by the two micrometers (M1 and M2) for when the west flange boundary end is in tension (position D−) is that a flexural crack has formed and runs through the application points of the two micrometers along the inside of the flange. Photos from the experiment support this (Figure 6b), where it is possible that the concrete surface crack affects the attachment of the micrometer to the wall, causing some small uplift. As discussed in the previous paragraph for UW2, another possible explanation for these discrepancies is the obvious increase in vertical deformation as measured by the micrometer if a strain gradient across the thickness is present, which was likely, due to the imposed rotation to the wall. For example, comparing the measured strain profiles for the interior and exterior rebars from Ch4 of the DFOS for wall unit UW3 confirms that a strain gradient was present (e.g., Figure 9a). The offset distance of the micrometer from the wall surface would inherently amplify the actual anchorage slip in the presence of strain gradients through the wall thickness. Furthermore, potential slip between the tip of the micrometer and the glass plate (see Figure 6b), which was glued to the metal plate for attachment to the wall, would only have exacerbated this behavior. On closer inspection, the glass plates were not used to support the tip of these micrometers for testing unit UW1 (i.e., compare Figure 6a with Figure 6b). While there were less discrepancies between the comparisons of the micrometer measurements and the calculated DFOS anchorage slip, the loading imposed to UW1 meant that there was negligible strain gradient through the thickness, which also made comparisons of these instruments more favorable. Another observation in Figure 16a is the larger Δ*_v_* as measured by micrometer M5 in comparison to the DFOS at position D- (i.e., LS 10, LS 13, LS 16, etc.). One possible explanation for this was provided in the previous paragraph regarding the same observation with unit UW2 (i.e., the twisting applied causes the neutral axis to be skewed diagonally across the thickness of the wall). One last curious observation is the increasing residual vertical deformation by M5 after LS 18 (Figure 16a), which results in a larger measured anchorage slip by the micrometer in comparison to that calculated by the DFOS. The DFOS strains corresponding to the interior rebar in this region confirm that, at the beginning of LS 18, the rebars were performing pre-yield (max strain of 2.364 mm/m). Thus, the residual deformation is unlikely to be a result of inelastic behavior from the reinforcing steel. Instead, it is again possible that flexural cracks have formed close to the fixation of the micrometer on the surface of the wall. If these cracks cannot close all the way, it is possible that the micrometer does not return to an anchorage slip measurement (Δ*_v_*) of zero on return to centering the wall (at position O). 

## 4. Conclusions

The dataset from a recent experimental program testing three large-scale reinforced concrete (RC) U-shaped walls provided an unusual opportunity to compare surface instrumentation (e.g., LVDTs, DIC, and micrometers) with sensors internal to the structural member using Distributed Fiber-Optic Sensors (DFOS). A summary of some of the key findings from this research investigation is given in the paragraphs below. They are relevant to the extent that, up to the present time, refined strain measurements in the core of RC members (either in the rebars or in the concrete itself) were extremely challenging to obtain reliably. However, the latter are key, as they ultimately govern the member response and allow to evaluate the accuracy of the modelling hypotheses behind many theories of structural mechanics, and hint toward their future improvement. Since most past instrumentation monitored member surface displacements, the present work offers a critical inspection of such techniques, as well as several recommendations.

Strain profiles derived from all surface instrumentation used in this experiment, including LVDTs, DIC, and micrometers, are unable to account for the strain penetration length into the foundation. Considering that previous RC wall tests in laboratories have used rudimentary spot sensors on the surface, such as LVDTs, to estimate average base strains, it is inevitable that strain and curvature demands at the wall base have been under or overestimated, depending on the empirical assumptions possibly considered for the strain penetration length. Conversely, the measurements from the DFOS are able to provide high-resolution data related to the strain penetration into the foundation of the three wall units, and how it evolves with the type of loading and ductility demand.

The different loading regimes imposed to the three wall units showed the importance of surface sensor placement. For example, wall unit UW2 was subjected to a reverse-cyclic torsional rotation, where a strain gradient across the thickness of the flanges, due to the combination of warping and circulatory torsion, was observed. While the DFOS bonded to the two rebars across the thickness of the wall was able to measure the difference in the strain profiles, a single array of LVDTs was not able to observe any strain gradient effects across the thickness.

Using the DFOS and LVDT strain profiles as a reference, the high-to-medium resolution DIC strains were corrected using a moving average to provide more coherent values, usable for engineering practice. The peaks of large tensile and compressive strains derived from the high-resolution DIC data were found to correlate well to the spacing of the flexural cracks and transverse reinforcement placement. In this study, a moving average of the uncorrected (i.e., raw) DIC data using a base length of 200 mm was found to provide better estimates of the strain profiles in comparison to the DFOS profiles. However, it is not known whether this base length can be used for processed DIC strain fields of all RC walls or other structures, and may be dependent on other factors. Therefore, since DIC techniques are now widely used in the large-scale testing of RC structures, more research in this area is needed.

Some discrepancies were also observed when evaluating the vertical displacement measured by micrometers at the base of the walls in comparison to those calculated with the DFOS strains. These discrepancies are again mainly attributed to difficulties associated with the micrometer measurements, whereas the computations from DFOS are judged to be reliable. It is postulated that some of the tensile displacement discrepancies observed are due to surface cracking of the concrete affecting the fixation of the micrometers to the surface of the walls.

Based on the above findings and observations, some summary recommendations are provided: (i) Spot sensors (e.g., LVDTs, micrometers, lasers, etc.) should be placed carefully with explicit considerations of the type of loading applied to the structure. For example, the placement of spot sensors should avoid positions of transverse reinforcement for flexurally governed RC structures, where it was shown in this research, using the DFOS strain profiles, that cracks are likely to form at the locations of these rebar placements; (ii) Future research should aim at evaluating this data, as well as other data, specifically to investigate the strain penetration length of RC walls; and (iii) It is recommended that high-resolution DIC strain fields are processed using a moving average with a base length of at least 200 mm to derive more reasonable estimates of the longitudinal strains in RC structures.

While there are some clear advantages to using DFOS to measure reinforcement strains of RC structures, some disadvantages were also observed during testing. The most obvious of the shortcomings is the current strain measurement limitation of approximately 10–15%. For flexurally-governed well-detailed RC specimens, much larger strains are to be expected, and therefore, this strain limitation currently imposed by the software will need to be overcome in the future for DFOS to have large application potential in this area of research. It is noted that the observed localized peaks in tensile strains may have resulted in the debonding of the fiber from the rebar, followed by possible buckling upon load reversal, and consequent false readings of net positive (tensile) strain in this localized region. Naturally, this latter observation is also dependent on how the fibers are bonded to the rebar, and further research is needed to find better approaches for extended measurement reliability. Thermal effects to the DFOS from the rebar behaving inelastically could also explain the net-tension strain observed, which will be explored by the authors in future coupon tests.

## Figures and Tables

**Figure 1 sensors-23-01745-f001:**
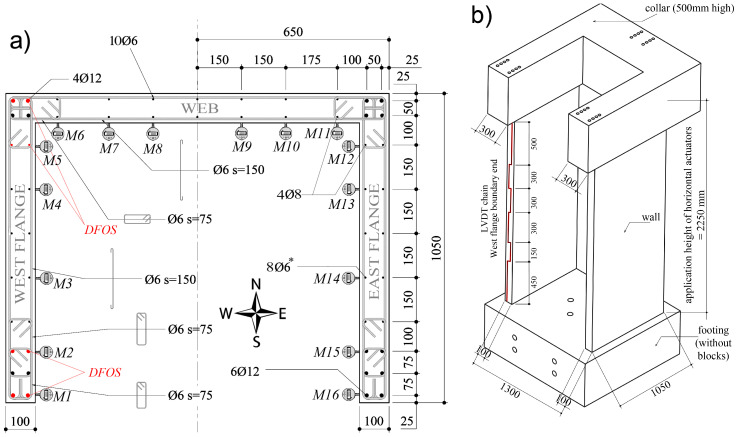
(**a**) Cross-section and reinforcement layout with indication of DFOS rebars and micrometer instruments (M1–M16, clockwise from boundary end of west flange); and (**b**) Elevation view of wall units with indication of the LVDT chain on the west flange boundary end.

**Figure 2 sensors-23-01745-f002:**
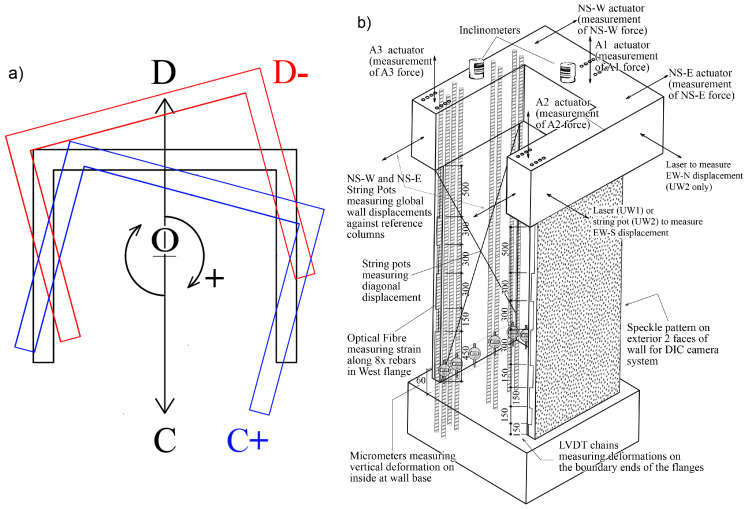
(**a**) Wall cross-section with the different loading positions; and (**b**) The different measurement instrumentation devices (not to scale; dimensions in mm).

**Figure 3 sensors-23-01745-f003:**
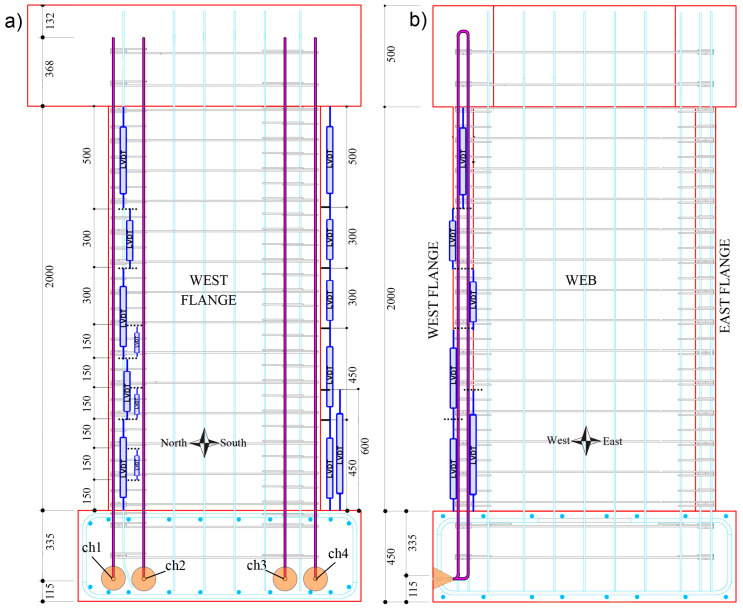
Location of the longitudinal rebars instrumented with fiber-optic sensors (purple, full opacity rebars): (**a**) Elevation view of the wall from the west with fiber-optic channels (“ch”) 1, 2, 3, and 4; and (**b**) Elevation view of the wall from the south. The LVDT chain to the south, along the boundary end of the west flange, and to the north, along the corner of the west flange-web intersection, is also depicted.

**Figure 4 sensors-23-01745-f004:**
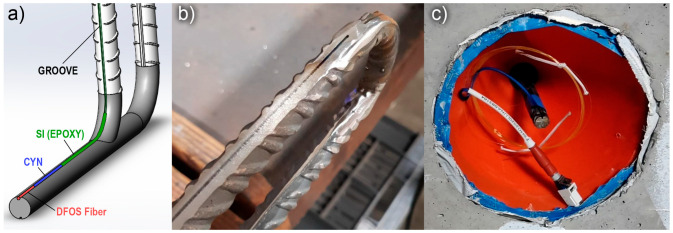
(**a**) Graphical representation of the bonding technique to the rebars and welded segment used in the foundation; (**b**) Groove made along the longitudinal rebar, showing the rounded and welded segment used in the head to join the two parallel rebar layers; and (**c**) the connector and termination of the fiber-optic cable was protected in the concrete footing using a plastic orange cone.

**Figure 5 sensors-23-01745-f005:**
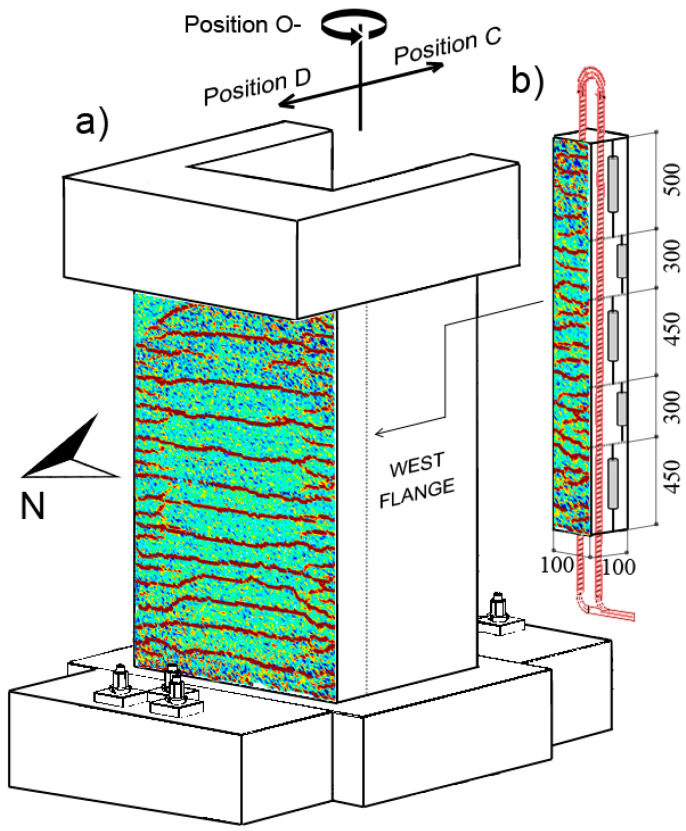
(**a**) 3-dimensional representation of the RC U-shaped wall units with foundation blocks and DIC strain results on the surface of the web; and (**b**) Corner-region “column” of the west flange-web intersection (100 × 100 mm^2^ cross-section) with DIC surface, embedded DFOS sensors (Channel 1), and LVDT chain, used in the current research.

**Figure 6 sensors-23-01745-f006:**
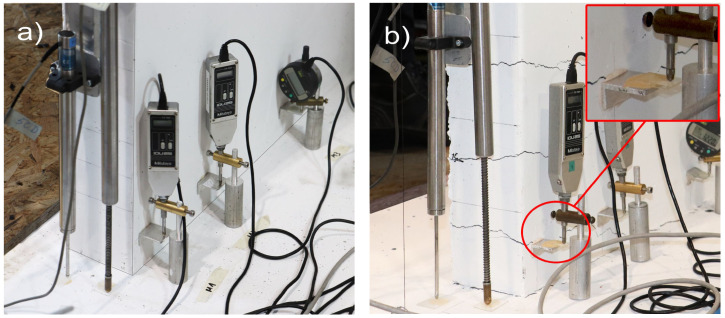
Photos of the inside edge of the west flange with micrometers (M1, M2, and M3) and base LVDTs attached close to the exterior and interior faces of the flange boundary end (**a**) Before testing of UW1; and (**b**) At LS34 of UW3 (position C+) showing flexural-shear crack running through attachment point of M1.

**Figure 7 sensors-23-01745-f007:**
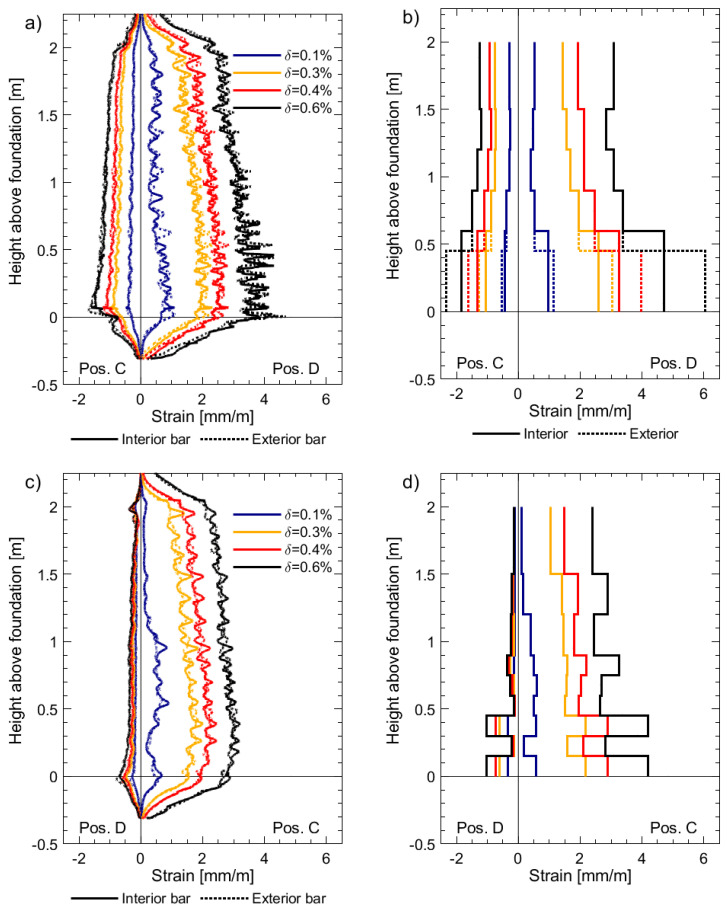
Longitudinal strain profiles for unit UW1 subjected to pure flexure at different imposed drifts: (**a**) Channel 4 of the DFOS; (**b**) chain of LVDTs located on the west flange boundary end; (**c**) Channel 1 of the DFOS; and (**d**) chain of LVDTs located on the west flange-web intersection.

**Figure 8 sensors-23-01745-f008:**
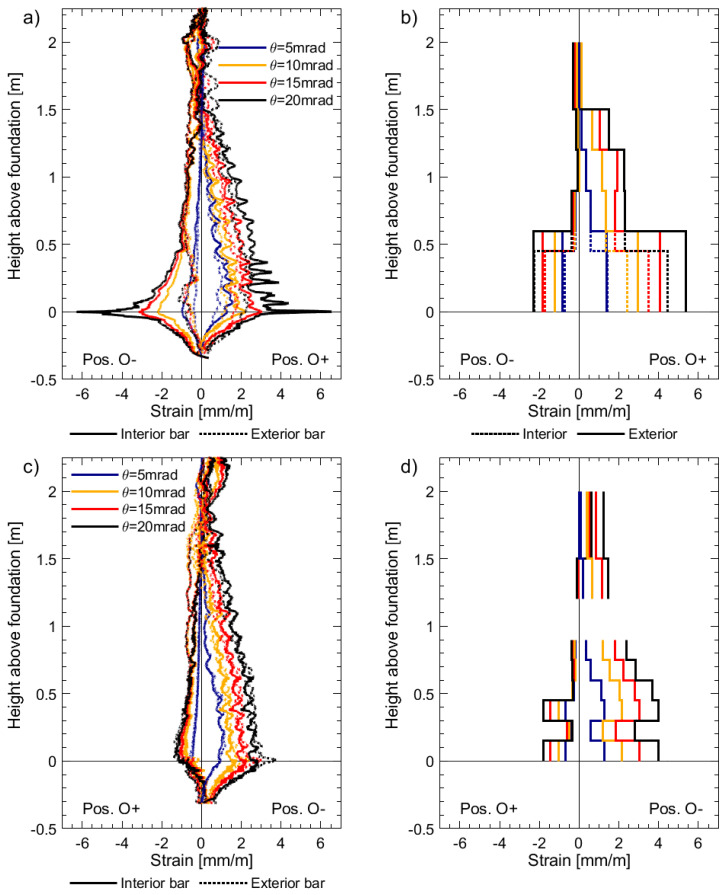
Longitudinal strain profiles for unit UW2 subjected to pure torsion at different imposed twists: (**a**) Channel 4 of the DFOS; (**b**) Chain of LVDTs located on the west flange boundary end; (**c**) Channel 1 of the DFOS; and (**d**) Chain of LVDTs located on the west flange-web intersection.

**Figure 9 sensors-23-01745-f009:**
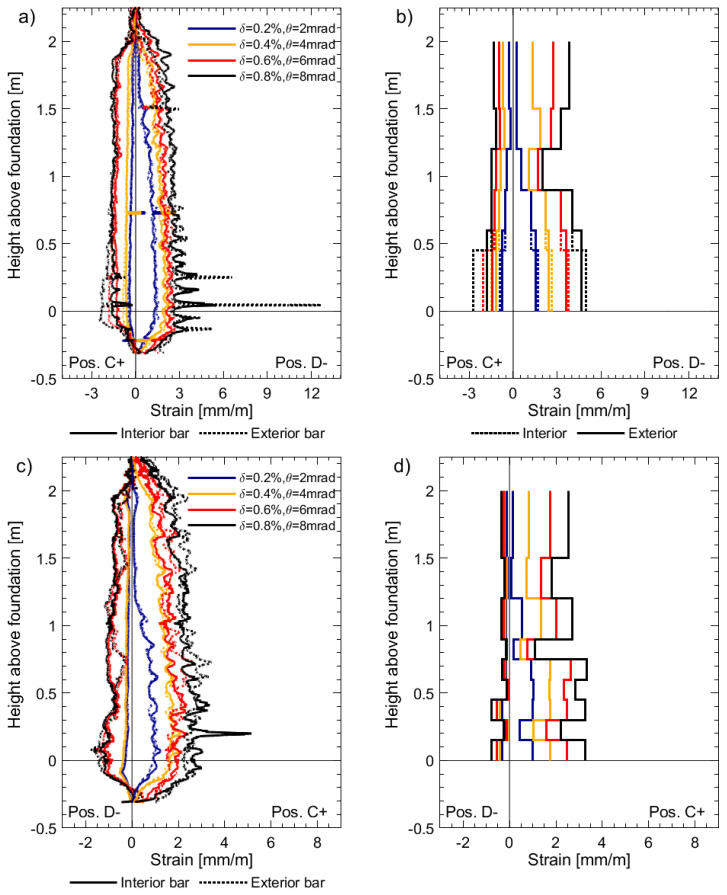
Longitudinal strain profiles for unit UW3 subjected to a combination of flexure and torsion: (**a**) Channel 4 of the DFOS; (**b**) Chain of LVDTs located on the west flange boundary end; (**c**) Channel 1 of the DFOS; and (**d**) Chain of LVDTs located on the west flange-web intersection.

**Figure 10 sensors-23-01745-f010:**
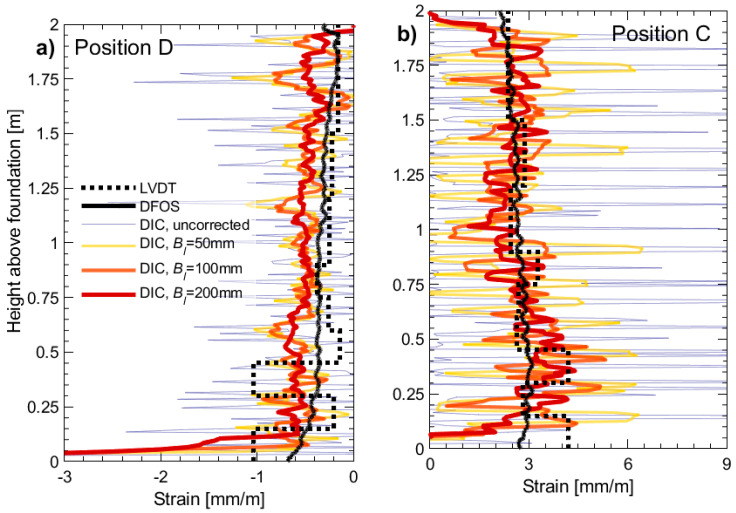
DIC strains profiles for the web-West flange intersection of wall unit UW1 at: (**a**) Position D (compression); and (**b**) Position C (tension) at *δ* = 0.6%. A moving average was used to smooth the DIC profiles using different span base lengths (*B_l_* = 50 mm, 100 mm, and 200 mm). The strain profiles are compared to the profiles measured from the average of the DFOS strain profiles (black solid lines) and LVDT strains (black dashed lines).

**Figure 11 sensors-23-01745-f011:**
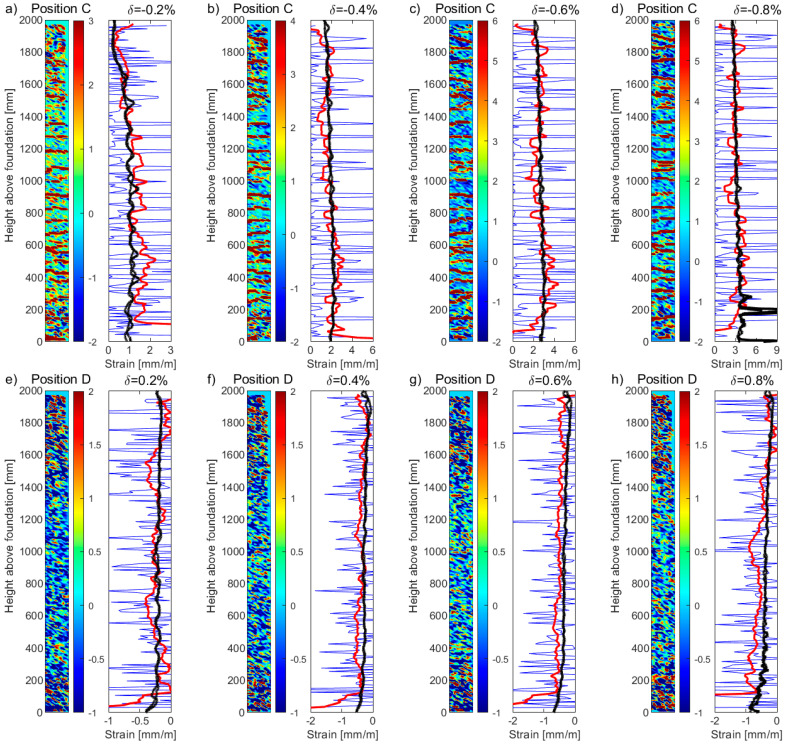
DIC and DFOS longitudinal strain profiles for the west web-flange intersection of wall unit UW1 subjected to flexure, in tension (Position C): (**a**) *δ* = −0.2%; (**b**) *δ* = −0.4%; (**c**) *δ* = −0.6%; (**d**) *δ* = −0.8%, and in compression (Position D); (**e**) *δ* = 0.2%; (**f**) *δ* = 0.4%; (**g**) *δ* = 0.6%; and (**h**) *δ* = 0.8%. The thin blue lines are the uncorrected strains determined from the DIC data, whereas the thick red lines are the corrected strains (i.e., moving average over 200 mm) determined from the DIC data. The solid black lines are the strain measurements from the DFOS (i.e., 2 × *Φ*12 rebars, Channel 1). A heatmap is provided next to each plot, representing the uncorrected, raw DIC strain (in units of mm/m).

**Figure 12 sensors-23-01745-f012:**
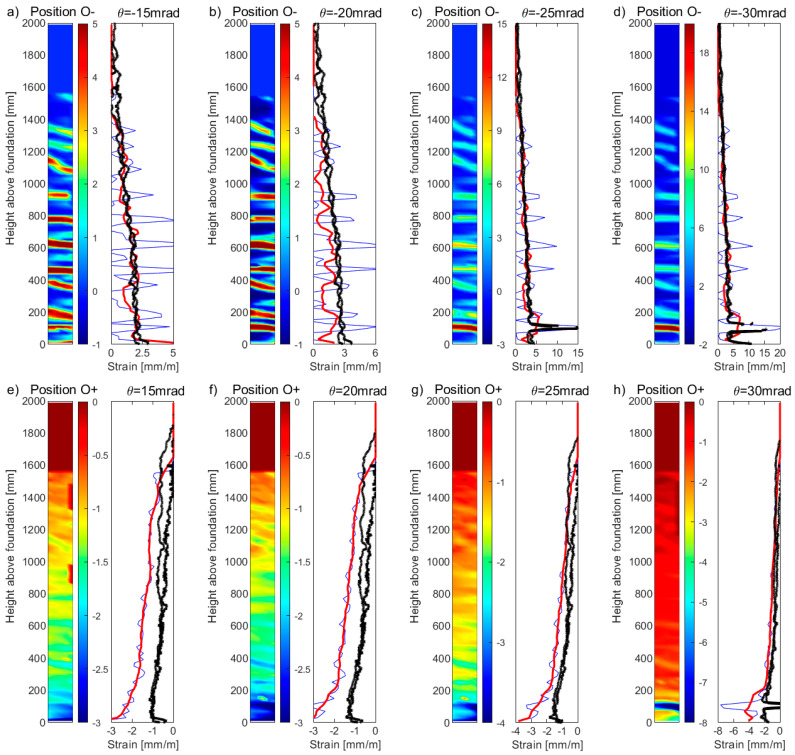
DIC and DFOS longitudinal strain profiles for the west web-flange intersection of wall unit UW2 subjected to torsion, in tension (Position O−): (**a**) *θ* = −15 mrad; (**b**) *θ* = −20 mrad; (**c**) *θ* = −25 mrad; (**d**) *θ* = −30 mrad, and in compression (Position O+); (**e**) *θ* = 15 mrad; (**f**) *θ* = 20 mrad; (**g**) *θ* = 25 mrad; and (**h**) *θ* = 30 mrad. The thin blue lines are the uncorrected strains determined from the DIC data, whereas the thick red lines are the corrected strains (i.e., moving average over 200 mm) determined from the DIC data. The solid black lines are the strain measurements from the DFOS (i.e., 2 × *Φ*12 rebars, Channel 1). A heatmap is provided next to each plot, representing the uncorrected, raw DIC strain (in units of mm/m).

**Figure 13 sensors-23-01745-f013:**
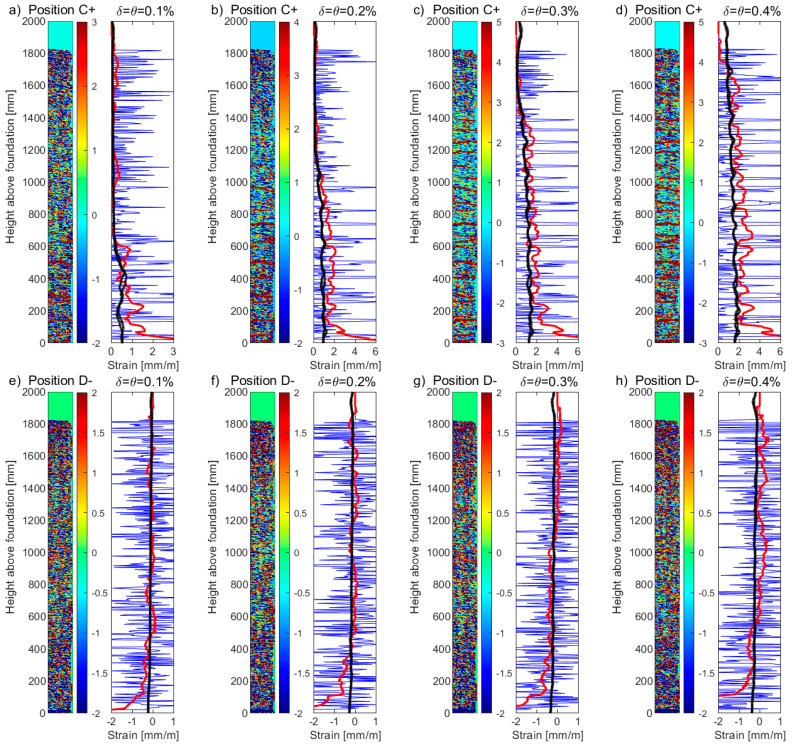
DIC and DFOS longitudinal strain profiles for the west web-flange intersection of wall unit UW3 subjected to flexure and torsion, in tension (Position C+): (**a**) *δ* = −0.1%, *θ* = 1 mrad; (**b**) *δ* = −0.2%, *θ* = 2 mrad; (**c**) *δ* = −0.3%, *θ* = 3 mrad; (**d**) *δ* = −0.4%, *θ* = 4 mrad and in compression (Position D); (**e**) *δ* = 0.1%, *θ* = −1 mrad; (**f**) *δ* = 0.2%, *θ* = −2 mrad; (**g**) *δ* = 0.3%, *θ* = −3 mrad; and (**h**) *δ* = 0.4%, *θ* = −4 mrad. The blue thin lines are the uncorrected strains determined from the DIC data, whereas the red thick lines are the corrected strains (i.e., moving average over 200 mm) determined from the DIC data. The solid black lines are the strain measurements from the DFOS (i.e., 2 × *Φ*12 rebars, Channel 1). A heatmap is provided next to each plot, representing the uncorrected, raw DIC strain (in units of mm/m).

**Figure 14 sensors-23-01745-f014:**
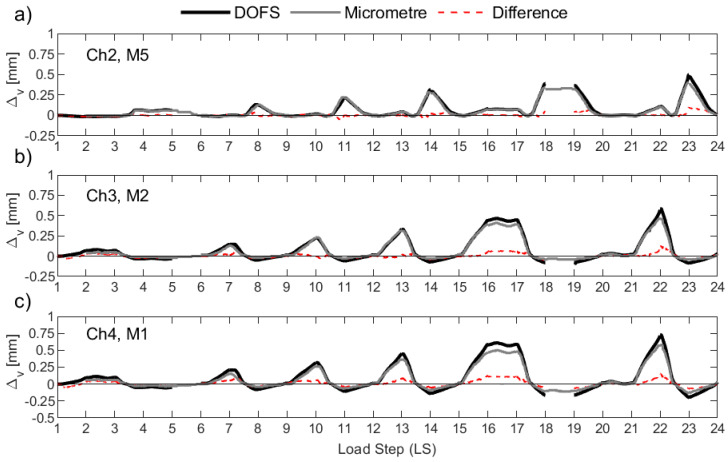
Anchorage slip (Δ*_v_*) calculated from the distributed optical fiber sensor (DFOS) strain profiles compared to the micrometers for unit UW1 subjected to in-plane flexure: (**a**) DFOS Channel 2, Micrometer 5; (**b**) DFOS Channel 3, Micrometer 2; and (**c**) DFOS Channel 4, Micrometer 1.

**Figure 15 sensors-23-01745-f015:**
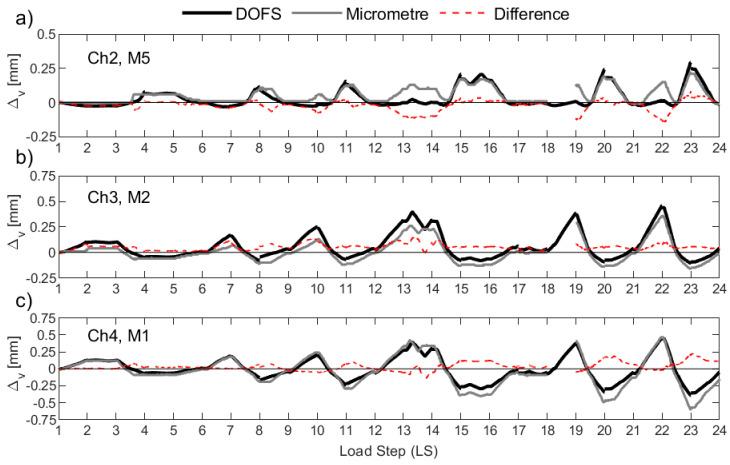
Anchorage slip (Δ*_v_*) calculated from the distributed optical fiber sensor (DFOS) strain profiles compared to the micrometers for unit UW2 subjected to torsion: (**a**) DFOS Channel 2, Micrometer 5; (**b**) DFOS Channel 3, Micrometer 2; and (**c**) DFOS Channel 4, Micrometer 1.

**Figure 16 sensors-23-01745-f016:**
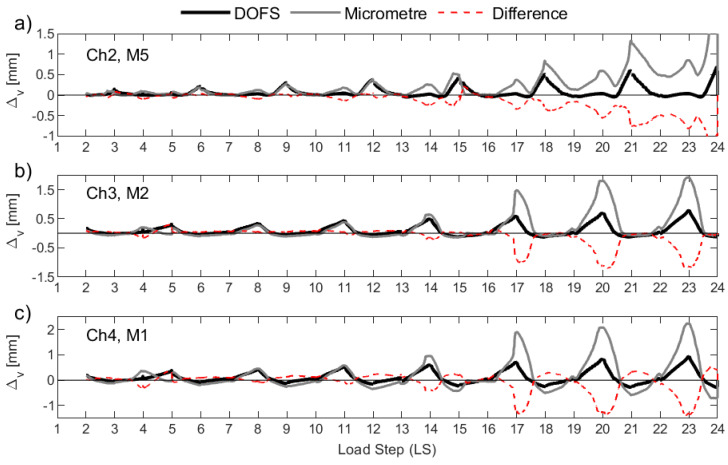
Anchorage slip (Δ*_v_*) calculated from the distributed optical fiber sensor (DFOS) strain profiles compared to the micrometers for unit UW3 subjected to torsion and flexure: (**a**) DFOS Channel 2, Micrometer 5; (**b**) DFOS Channel 3, Micrometer 2; and (**c**) DFOS Channel 4, Micrometer 1.

**Table 1 sensors-23-01745-t001:** Correlation parameters for the processed DIC files for each wall unit.

	UW1	UW2	UW3
Facet Size (px)	19	99	19
Accuracy (px)	0.5	0.1	0.5
Residuum (gray)	20	20	20
3D Residuum (px)	1	2	1
Grid Spacing (px)	10	35	10
Horizontal Grid Spacing (mm)	10	20	5
Vertical Grid Spacing (mm)	10	20	5

## Data Availability

The original experimental dataset that was used to undertake the research investigation in this paper can be downloaded from the publicly accessible platform *Dataverse*, doi: https://doi.org/10.14428/DVN/FDJ4EU, accessed on 3 November 2022.

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
