# Peer review of "Core versus Surface Sensors for Reinforced Concrete Structures: A Comparison of Fiber-Optic Strain Sensing to Conventional Instrumentation"

_sensors, 2023, doi:10.3390/s23031745_

Round 1

Reviewer 1 Report

Your paper presents some interesting results although at the moment some improvements are required before I can recommend publication:

1. As Pascal said "I have made this longer than usual because I have not had time to make it shorter." In general this paper could be much more concisely written. But since I do not believe the journal has a word limit, the only place I would really like to see tightened up are the conclusions, which really do make it difficult to see the importance of the paper for lack of conciseness.

2. 23: providing keywords that are essentially the same is confusing. Please use DFOS or DOFS but not both. Also, is LVDT really a meaningful keyword? Similarly "bars" needs a bit more context, I'm assuming reinforcement bars? Although I would also question it as a keyword. Also, "fibre" needs more context, "fibre optic cable"?

3. 34: does Scot not spelled Scott? Check and revise as necessary.

4. 68-70: rather than referring to "low" and "high" cost it would be more valuable to quantify these values.

5. 99-100: it seems the authors have published on this work, granted in conferences but the papers are available. As such, a brief statement acknowledging that work and indicating how the current work is different is required.

6. 108: in the references section it says this paper is submitted not published. Submitted papers cannot be cited as there is no guarantee they will accepted. If the reference is incorrect and the paper is accepted please adjust the reference. If it is a submitted paper it should not be referenced.

7. 121-123: please refer to a figure that shows the instrumented bar locations (I believe Fig. 1 does but if there is no such figure one should be added).

8. 126: this statement makes the reader question the value of this data. Please include a statement that indicates why this data is representative rather than the current explanation which just seems to be "we had this data and didn't want to waste it".

9. 182-183: I suspect DIMIONE systems is just a reseller / agent for Luna and if that's the case they should not be mentioned in a technical paper. Only the equipment manufacturer should be mentioned. Also, saying the Luna system was used to "compile" the data is hugely misleading as it makes it sound like it is a DA. It is a critical part of the sensor system so this should be rephrased to be more technically sound.

10. 287: please present figures in order (i.e. do not introduce Figure 6 before Figure 5).

11. 342: should be "Photos" rather than "Photo"

12. 361-367: the reason for this difference in measurements is not obvious and should be discussed further. Though locally the surface and rebar strains can be quite different. On average they should be the same to ensure compatibility. So why these are so different over a reasonably long gauge length is not clear. You seem to suggest later on that it has to do with strain penetration but this explanation is not clear (especially for the broader audience of a sensors journal).

13. Figure 7, 8, and 9: the challenge with these figures is that you are expecting the reader to be able to do the comparison from one figure to another (a versus b), which is not very effective. Can you instead use fewer displacement ratios and plot the data on one figure? Since the goal of your paper (I assume) is not to talk about wall behavior, the fact that you would show fewer displacement steps but a better comparison of the sensor data can't hurt. Then see comment 12 as at the moment I do not feel this discussion of the differences between these two sensor systems is strong. But because the data is not presented in a way where the reader can effectively see the comparison it is also hard to judge the quality of the comparison.

14. 487: what does the phrase "reasonable strains" mean? What do you consider reasonable and why? The phrase is not effective for technical writing and should be refined ideally with some form of quantification.

15. 537: here again what is "reasonable"? In fact, here it is much more important as this is a paper on sensors and based on Figure 10, even with a 200 mm average the DIC data does not appear to visually match the DFOS data. So, this comes across as a hand wavy attempt to say that the data is good enough. But is it? Without any idea of what you think is reasonable with some level of quantification it really doesn't seem good enough. It is necessary to have some discussion of % errors or other statistical means of comparing the data and suggesting what a limit on reasonable is.

16. 582: as per my previous comment, this isn't really a "reasonable" level of analysis when comparing sensor systems for a Sensors journal.

17. Figure 11, 12, and 13: what is the contour plot and the scale on the legend for the contour plot?

Reviewer 2 Report

The paper entitled “Core versus surface sensors for reinforced concrete structures: a comparison of fiber-optic strain sensing to conventional instrumentation” falls within the scope of this Journal. The presented study is interesting and novel. However, some revisions are required before it can be considered for publication. The following comments should be addressed to improve the technical quality of the manuscript.

1.      All fiber optic sensors (FOS) were widely known to have disadvantages of temperature dependency including DFOS. Therefore, how the authors eliminate the temperature effects on the DFOS?

2.      How many times the experiments (flexure, torsion, and a combination of flexure and torsion) were repeated for each specimen? Perhaps the authors should repeat at least 3 times the measurements for each specimen. This is to evaluate the repeatability of the proposed sensor.

Thank you. I look forward to your kind response.

Reviewer 3 Report

I believe that the work submitted for consideration is a completed scientific study and has a high practical significance. It is well arranged, the experiment is described in detail and can be reproduced by other researchers. However, it is worth highlighting a few minor remarks that should be corrected before publication in the journal:
- In the introduction, the authors note that the risk of damage to point sensors increases if they are located near a crack. But at the same time, it is known that with an increase in the crack, an optical fiber can also be destroyed. I recommend rephrasing the sentence.
- in the description of the experiment it is indicated that the data obtained from distributed fiber-optic sensors were averaged by a sliding window of 10 mm in size, while another value appears in the conclusions: 200 mm. It is required to eliminate this contradiction.
- the authors talk about the "general" limitations of distributed fiber-optic sensors in terms of the impossibility of finding the peaks of the cross-correlation functions of the spectra at large deformations, while operating only with the tools of optical reflectometry in the frequency domain. It is necessary to explain why the research team ignores the technology of Brillouin reflectometry or Brillouin analysis - BOTDR / BOTDA. At the same time, this technique is successfully used in SHM, and its performance is improved by data processing algorithms and artificial intelligence, which makes it possible to use the sensor in extreme conditions. This type of sensor can have high length resolution [10.1364/OL.424856] and good sensitivity [10.3390/s22072677].
- when using the method of optical reflectometry in the frequency domain, the frequency coordinate of the peak depends not only on deformation, but also on temperature. The authors do not indicate the method by which they excluded the effect of temperature on the experiment.
- to denoise the trace, the authors smooth the obtained data by averaging. At the same time, distributed sensor settings such as sweeping range, sampling parameters, and others can bring success at an earlier stage.

Round 2

Reviewer 1 Report

Thank you for addressing most of my comments.

Author Response

The authors again thank the reviewer for their time. The authors have gone through the manuscript again and conducted a spell check, and correspondingly made some very minor changes. A pdf of the revised manuscript is attached with some highlighted changes. The word doc of the revised manuscript is clean and does not have these areas highlighted.
